# Quick mining in dense data: applying probabilistic support prediction in depth-first order

Muhammad Sadeequllah[1], Azhar Rauf[1], Saif Ur Rehman[1] and Noha Alnazzawi[2]

[1] Department of Computer Science, University of Peshawar, Peshawar, KP, Pakistan
[2] Computer Science and Engineering Department, Yanbu Industrial College, Yanbu, Saudi Arabia



Corresponding author
Muhammad Sadeequllah,
sadeequllah@uop.edu.pk

## ABSTRACT

Frequent itemset mining (FIM) is a major component in association rule mining, significantly influencing its performance. FIM is a computationally intensive nondeterministic polynomial time (NP)-hard problem. At the core of FIM is the task of computing support of candidate itemsets. This problem becomes more severe when the dataset is dense as the support is computed for millions, or even billions, of candidate itemsets. The rapid growth of data further exacerbates this problem. To achieve high scalability and efficiency, recently, researchers have proposed various approaches to approximate the support of an itemset using as small a subset of transaction data as possible. In addition to efficiency, accuracy is another important metric for these algorithms. They strive to increase true positives and reduce false negatives and false positives. One such recently proposed approximate FIM algorithm is Probabilistic Breadth-First (ProbBF), which is highly efficient for dense data due to its unique approach of not using transactional data beyond 2-size itemsets. Unlike other counterparts, this algorithm requires no additional input parameters beyond the traditional support threshold. However, ProbBF is a breadth-first algorithm, and it is well-established that breadth-first FIM algorithms consume significantly more memory than depth-first algorithms on dense datasets. It is also worth noting that significantly high memory consumption slows run-time performance of an algorithm due to low utilization of locality of reference, thrashing, and aggressive garbage collection *etc*. This article proposes a FIM algorithm, ProbDF, that discards transaction data after determining all frequent itemsets of sizes one and two. For frequent itemsets of size three or more, it employs a probabilistic support prediction model (PSPM) to predict their support probabilistically. PSPM, first proposed with ProbBF, uses lightweight calculations that exclude transaction data. Our experiments demonstrate that ProbDF, with its depth-first search strategy tailored to PSPM and other optimizations, is efficient in terms of time and space, and successfully generates the majority of frequent itemsets on real-world benchmark datasets. However, due to the probabilistic nature of ProbDF, some compromise in quality is inevitable.

## INTRODUCTION

Frequent itemset mining (FIM) is the fundamental task in Association rule mining that has diverse applications. Some of these applications are market basket analysis, consecutive clicks by users in click stream mining, stock analysis, business decisions, user actions in intrusion detection, medical diagnosis, and bio-informatics. The concept of FIM, first presented in *Agrawal et al. (1996)*, represents an unsupervised data mining approach aimed at discovering frequently co-occurring items within transaction datasets. This technique entails substantial computational and memory demands, as it explores all potential combinations of frequent items present in the transaction dataset. The most resource-intensive aspect of an FIM algorithm lies in the support counting of itemsets derived from transaction data.

The frequent itemsets generation process involves identifying sets of items that appear together frequently in a transaction dataset. It starts with calculating the support (frequency) of individual items, retaining those that meet a predefined minimum support threshold. Next, it generates candidate itemsets of size 2 by combining frequent items, calculates their support, and prunes those that are not frequent. This process iteratively continues for larger itemsets, using the Apriori property, which ensures that only those candidate itemsets whose all subsets are frequent are considered. The process terminates with a collection of itemsets that frequently appear together in the transactions.

### Motivation

Frequent itemset mining is a highly computational and memory-intensive task. It needs to find all the frequent itemsets and their possible combinations that are also frequent. FIM also needs to store the transaction data in memory associated with these discovered frequent itemsets. This problem becomes even more severe when FIM is deployed to large dense datasets. A dense dataset is one in which every frequent item is assumed to be present in almost every transaction (*Fournier-Viger et al., 2017*; *Luna, Fournier-Viger & Ventura, 2019*). One reason for this is that the number of frequent itemsets reaches into billions even for an average-sized dense dataset with only a few dozen frequent items.

To make FIM algorithms more efficient and scalable, different researchers have proposed various methods. One of these methods was to propose efficient data structures to speed up FIM on dense datasets, but the constant challenge introduced by advances in computing technology is the tremendous growth in data. Employing maximal frequent itemsets (MFI) mining in dense datasets is another solution. While the support of an itemset indicates its importance in the FIM domain, with the MFIs, we can only identify all the frequent itemsets–we cannot rank them based on their support because we lack the support information for each individual frequent itemset.

Some researchers have attempted to restrict the use of transactional data and calculate an approximate value for the support of itemsets. Though the downside of these algorithms is that they do not guarantee to generate all possible frequent itemsets, the resulting loss is marginal. These algorithms have employed different techniques to limit the transactional data used to derive the support of an itemset. For example, some researchers have used sampling techniques (*Li et al., 2016*), hashing methods

(*Zhang et al., 2018*), filtering techniques (*Abbasi & Moieni, 2021*), and clustering algorithms (*Fatemi et al., 2021*).

In one such recent effort, a probability-based technique, called probabilistic support prediction model (PSPM), was proposed in *Sadeequllah et al. (2024)*. This technique is used to probabilistically predict the support of an itemset. This article also proposes a FIM algorithm, called Probabilistic Breadth-First (ProbBF), that incorporates the PSPM technique. This algorithm is different from other FIM algorithms in that it does not use any additional parameter other than the minimum support threshold. Additionally, it uses zero transaction data in its support approximation. These characteristics render the ProbBF algorithm more efficient than other FIM algorithms, both approximate and exact. It is also worth noting that ProbBF is the first approximate FIM algorithm designed for dense data, whereas other approximate algorithms typically perform well with sparse data only.

However, problem with the ProbBF algorithm is that it is a breadth-first algorithm, and it is established that breadth-first algorithms are less efficient and less scalable than depth-first algorithms when deployed to dense data. This is due to the fact that the memory requirements of breadth-first and depth-first algorithms are respectively $O(w)$ (where $w$ is the maximum width of the search space tree) and $O(h)$ (where $h$ is the maximum depth of the search space tree). For dense data, $w$ is much more larger (order of magnitudes) than $h$. High memory requirements could also slow an algorithm due to the reasons such as low utilization of locality of reference, thrashing, and aggressive garbage collection. Although, a pure depth-first algorithm also suffers from slow runtime due to large number of candidate generation. A better compromise is a hybrid search strategy used by many breadth-first FIM algorithms (*e.g.*, MAFIA (*Burdick et al., 2005*), Genmax (*Gouda & Zaki, 2005*), NegFIN (*Aryabarzan, Minaei-Bidgoli & Teshnehlab, 2018*), SelPMiner (*Bai et al., 2019*)).

Another important shortcoming of the ProbBF algorithm is its lack of support for additional pruning techniques beyond apriori pruning. Many modern FIM algorithms designed for dense datasets incorporate a highly efficient pruning technique initially used by Maxminer and later known by various names such as Parent Equivalence Pruning (PEP) in MAFIA (*Burdick et al., 2005*), Genmax (*Gouda & Zaki, 2005*), and SelPMiner (*Bai et al., 2019*), Hypercube Decomposition in LCM (*Uno, Kiyomi & Arimura, 2004*), and Promotion Pruning in PrePost+ (*Halim, Ali & Khan, 2020*), dFIN (*Xun et al., 2021*), and negFIN (*Aryabarzan, Minaei-Bidgoli & Teshnehlab, 2018*). This technique prunes entire subtrees rooted at frequent itemsets whose support equals that of their parent frequent itemsets, significantly enhancing the runtime efficiency of FIM algorithms on dense data.

Furthermore, ProbBF employs a set-enumeration tree to store frequent itemsets, which is memory inefficient. The nodes in this tree store pointers to both their children and parent nodes, with the frequent prefix distributed across multiple nodes. A more memory-efficient technique is needed to improve memory utilization in the proposed algorithm.

The ProbDF algorithm, proposed in this article, is an efficient depth-first algorithm that addresses all the shortcomings of ProbBF (*Sadeequllah et al., 2024*). ProbDF utilizes a novel depth-first traversal method that meets the requirements of the PSPM technique, a

cornerstone of ProbBF. It identifies all frequent children itemsets of an itemset using a breadth-first approach and subsequently traverses these children itemsets in a depth-first manner. This algorithm also introduces a technique to incorporate the Promotion Pruning concept. Lastly, it proposes a single vector technique to store all frequent suffixes.

### Contributions

The main contributions of this article are as follows:

1) This article proposes a depth-first algorithm ProbDF with a novel search strategy that ensures the support of all $(k$-1$)$ itemsets of a $k$-itemset be already known before evaluating the support of the $k$-itemset. This is an important requirement of the PSPM model.

2) This algorithm introduces a time and space efficient single vector technique to store a common $k$-1 size prefex for all the $k$-size itemsets currently being explored. This is contrary to the other algorithms that store $k$-1 frequent items prefix with every $k$-size frequent itemset.

3) This algorithm also introduces the concept of promotion pruning to approximate FIM algorithms to enhance its memory and runtime efficiency.

The structure of the article proceeds as follows: "Related Work" offers a review of the most pertinent literature to our proposed solution in the field of FIM. "Problem Setting and Preliminaries" introduces preliminary concepts relevant to frequent itemset mining. "Probabilistic Support Prediction Model" outlines the PSPM technique discussed earlier. Following this, "The Proposed Algorithm, ProbDP" unveils our proposed algorithm, ProbDF. "Experimental Results" delves into a comprehensive analysis of ProbDF's performance against state-of-the-art FIM algorithms using real-world benchmark datasets. Lastly, "Future Work" provides the article's concluding remarks.

## RELATED WORK

Mining frequent itemsets is a highly complex computational task, often classified as an NP-hard problem. This is especially true when FIM deals with large, dense datasets. Given the resource-intensive nature of this task and the ever-expanding size of datasets, research interest in this domain is still alive. While early efforts primarily focused on the development of frequent itemset mining algorithms (*Halim, Ali & Khan, 2020*; *Xun et al., 2021*; *Ledmi, Zidat & Hamdi-Cherif, 2021*; *Ghosh et al., 2023*), the landscape has since evolved into other itemset mining algorithms as well, such as sequential patterns mining algorithms (*Song, Ye & Fournier-Viger, 2022*; *Tang et al., 2022*), data stream mining algorithms (*Xiao & Hu, 2020*; *Lu et al., 2020*), graph mining algorithms (*Nguyen et al., 2022*; *Preti, Morales & Riondato, 2023*), approximate frequent itemset mining in uncertain data (*Lin, Fu & Hsueh, 2022*; *Bashir & Lai, 2021*), and high utility frequent itemset mining algorithms (*Kumar & Singh, 2023*; *Hung, Tung & Vo, 2021*). In this section, we evaluate frequent itemset mining algorithms that are the most relevant to our proposed algorithm.

In the FIM algorithms, particular attention is directed towards the exact frequent itemset mining algorithms, which utilize the entirety of transaction data for the

identification of frequent itemsets. These algorithms are more precise and do not involve the risk of false negatives or false positives. Research efforts continue towards enhancing the efficiency of these algorithms, as evidenced by recent publications (*Jamsheela & Raju, 2023*; *Lu et al., 2023*; *Fang et al., 2023*; *Qu et al., 2020*).

Recently, renewed interest shown by scholars in another avenue of research that explores approaches aimed at enhancing efficiency and scalability of FIM algorithms by approximating itemset support using only a small subset of transaction data. Early explorations in this area include the introduction of sampling techniques into the field of frequent itemset mining (*Toivonen, 1996*). In sample-based frequent itemset mining algorithms, a subset is extracted from the large dataset, and frequent itemsets are exclusively identified within this subset. This approach tries to reduce false negatives by lowering the support threshold, while false positives are addressed through an additional iteration over the discovered itemsets. However, this extra step compromises computational efficiency, thus undermining the intended goal of the proposed method (*Aggarwal & Han, 2014*). Furthermore, the selection of the optimal sample size parameter further complicates the application of sample-based methods because it governs the tradeoff between accuracy and efficiency (*Wu et al., 2015*; *Riondato & Upfal, 2015*; *Zhang, Pedrycz & Huang, 2017*). *Li et al. (2016)* experimented with dynamic sample size, however, he also introduced two other parameters to tune the algorithm. Some researchers have also investigated the application of clustering algorithms in FIM, *e.g.*, *Ordonez (2009)*. Recently, *Fatemi et al. (2021)* used clustering to approximate Maximal Frequent Itemsets (MFI). However, the problem of additional parameters tuning persists with clustering-based approximate FIM algorithms as well. It requires tuning the number of clusters and the Centroid threshold parameters. Hash-function-based methods were also explored to estimate the intersection size of multiple sets. *Cohen et al. (2001)* used MinHash for frequent itemsets mining, employing the Jaccard similarity index to approximate the support of candidate itemsets. *Pagh, Stöckel & Woodruff (2014)* proposed a method based on one permutation MinHash to estimate intersection sizes. *Dasu et al. (2002)* developed an estimator using $k$-permutation Minwise hashing. *Bera & Pratap (2016)* suggested three Apriori variations using LSH for frequent itemset computation. Recently, *Zhang et al. (2018)* employed Minwise Hashing to estimate itemset support counts using reduced transaction sets. However, hashing techniques also require additional parameters ($K$ and $E$ values). They retain complete transaction data and, in certain instances, choose to calculate support directly rather than relying on approximations, especially when $E$ values are high. Additionally, they employ transaction list signatures to estimate itemset support counts, which are essentially condensed transaction lists based on parameter $K$. Moreover, these algorithms employ complex computations on transaction data, and their reliance on even limited transaction data hampers their efficiency and scalability, as a moderate-sized dense dataset also results in substantial computations. Furthermore, the need for additional parametric settings complicates their usage.

In a recent research effort, the ProbBF (*Sadeequllah et al., 2024*) approximate FIM algorithm was proposed. ProbBF is a probabilistic approach that approximates the support of an itemset with no additional parameters except the traditional minimum support

**Table 1 Analysis of different types of approximate FIM algorithms.**

| Agorithm | Require extra parametric input | Dense data friendly design | Use transaction data in support approximation | Store multiple frequent prefixes | Use additional pruning except Apriori | Search space traversal strategy |
|---|---|---|---|---|---|---|
| Sampling-based Algorithms | Yes | No | Yes | Yes | No | Use both depth/ Breadth-first |
| Hash-based algorithms | Yes | No | Yes | Yes | No | Depth-First |
| Clustering-based algorithms | Yes | Yes | Yes | Yes | No | Not applicable |
| ProbBF algorithm | No | Yes | No | Yes | No | Breadth-first |
| ProbDF algorithm | No | Yes | No | No | Yes | Depth-first |

threshold. Another important feature of this algorithm is that it does not use transactional data for support approximation—it only probabilistically predicts the support count. Additionally, this algorithm employs relatively simple computations compared to its other approximate counterparts.

However, it is worth noting that ProbBF operates as a breadth-first algorithm. Numerous studies have established that breadth-first FIM algorithms tend to be less efficient than their depth-first counterparts, both in terms of memory usage and running time (*Aggarwal & Han, 2014*). Moreover, depth-first algorithms have incorporated parent support equivalence pruning techniques to enhance the efficiency of the mining process (*Aryabarzan, Minaei-Bidgoli & Teshnehlab, 2018*; *Deng, 2016*). Table 1 provides a concise summary of the methodologies and drawbacks of all the algorithms discussed in this section. Additionally, various symbols and notations used throughout this article are summarized in the notation table presented at the end of the article.

## PROBLEM SETTING AND PRELIMINARIES

In this section, we begin by offering a precise definition of the FIM problem. Following this, we explore commonly used data formats in FIM literature.

### Problem statement and definitions

Consider a set of $n$ transaction items, denoted as $I = \{i_1, i_2, \ldots, i_n\}$, and a collection of $m$ transactions, denoted as $T = \{t_1, t_2, \ldots, t_m\}$, where each transaction $t_i$ is a subset of $I$. Let $X \subseteq I$ be an itemset of size $k$, referred to as a $k$-itemset. For example, the set $\{A, B, C\}$ is a 3-itemset, which can be abbreviated as $ABC$.

If $Y \subseteq T$ and for every $y \in Y$ and $X \subseteq y$, then the support of $X$, denoted $\sigma(X)$, is defined as $\sigma(X) = |Y|/|T|$, where $|Y|$ is the support count of $X$. This implies that $\sigma(X)$ is the probability that the itemset $X$ appears in a randomly selected transaction from $T$. An itemset $X$ is considered frequent if $\sigma(X) \geq min\_sup$, where $min\_sup$ is a user-defined threshold for support.

**Table 2 Horizontal *vs.* vertical data format.**

| Transaction ID | Items | Items | Transaction IDs |
|---|---|---|---|
| 1 | Z Y W | Z | 1, 2, 3, 4, 5, 6, 7, 9, 10 |
| 2 | Z Y X S | Y | 1, 2, 3, 5, 7, 8, 9, 10 |
| 3 | Z Y X T | X | 2, 3, 4, 5, 6, 7, 9 |
| 4 | Z X W S | W | 1, 4, 6, 7, 8, 10 |
| 5 | Z Y X T S | T | 3, 5, 6, 8, 9 |
| 6 | Z X W T | S | 2, 4, 5, 10 |
| 7 | Z Y X W | | |
| 8 | Y W T | | |
| 9 | Z Y X T | | |
| 10 | Z Y W S | | |

An association rule $A \Rightarrow B$ represents the probability that a random transaction $t \in T$ contains both itemsets $A$ and $B$, given that $A \in I$, $B \in I$, $A \neq \emptyset$, $B \neq \emptyset$, and $A \cap B = \emptyset$. The support for the rule $A \Rightarrow B$ is the joint probability of $A$ and $B$, denoted as $P(A \cap B)$, which is computed as $\sigma(AB)$. The confidence of the rule $A \Rightarrow B$ is the conditional probability $P(B|A)$, defined as $\sigma(AB)/\sigma(A)$.

In a standard association rule mining process, all potential rules meeting specified criteria—such as user-defined thresholds for support and confidence—are generated for a given dataset. The most computationally demanding aspect of this process is frequent itemset mining. Determining the frequency of an itemset $X$ involves inspecting all transactions containing $X$, known as the support count of $X$, which entails significant computational expenses. In this article, we use a technique presented in the following section to approximate this value.

### Transactions data format

The transaction dataset comprises transactions, each assigned a unique transaction ID (TID) along with associated transaction items. FIM algorithms utilize either horizontal or vertical formats of this data for support count calculations of itemsets. For instance, Table 2 illustrates a transaction dataset of 10 transactions, depicted in horizontal format on the left and vertical format on the right. Calculating the support count of an itemset in the horizontal format involves tallying all transactions that include the itemset. Conversely, in the vertical format, the intersection operation is applied to the TID sets of the items constituting the itemset, with the support count being the size of the resulting TID set. ProbDF is tailored to utilize arrays that exclusively store the counts of these TIDs for all itemsets of size two.

## PROBABILISTIC SUPPORT PREDICTION MODEL

In this section, we reproduce the PSPM proposed in *Sadeequllah et al. (2024)*. This model is used to predict the support count of a $k$-size itemsets for $k \geq 3$. This model is recursively

applied to find the support of higher order sets of size four and higher. This model is as follows:

Let $A$, $B$, and $C$ be any three finite sets, and suppose they are subsets of a universal set $U$. According to the probability theory:

$$P(A \cap B) = P(A) \times P(B|A). \tag{1}$$

If A and B are independent events then $P(B|A) = P(B)$ and (2) becomes:

$$P(A \cap B) = P(A) \times P(B). \tag{2}$$

Another way to write Eq. (1) is:

$$P(A \cap B) = P(A) \times P(B) \times C_f(A, B). \tag{3}$$

The $C_f$ shows the correlation of two probabilistic events $A$ and $B$, which is written as follows.

$$C_f = \frac{P(A \cap B)}{P(A) \times P(B)}. \tag{4}$$

The $C_f$ is called the correlation factor. If $A$ and $B$ are independent events then $C_f(A, B) = 1$. In this case Eq. (3) equals Eq. (2). Though Eqs. (1) and (3) are equal, the Eq. (3) has the advantage that it separates the data dependence. If $P(A \cap B)$, $P(A \cap C)$ and $C_f(B, C)$ are already known, and $X = (A \cap B)$, $Y = (A \cap C)$, it could be shown that:

$$P((X \cap Y)|A) = P(X|A) \times P(Y|A) \times C_f((X, Y)|A). \tag{5}$$

It is clear in Eq. (5) that $(X \cap Y) = (A \cap B \cap C)$, but $(A \cap B \cap C)$ is the quantity that is required. For the dense data, it was shown in *Sadeequllah et al. (2024)* that $S \rightarrow T$ more often if $S$ is the immediate proper subset of $T$. Since $X$, $Y$, and $A$ are the immediate proper subsets of $B$, $C$, and $U$ respectively, $X \rightarrow B$, $Y \rightarrow C$, and $A \rightarrow U$. Therefore, we assume that the correlation of $B$ and $C$ in $U$ approximates the correlation of $X$ and $Y$ in $A$ as $X$ and $Y$ are the immediate proper subsets of $A$. This is due to the fact that if $X \rightarrow B$, $Y \rightarrow C$, and $A \rightarrow U$ then $C_f((X, Y)|A) \cong C_f(B, C)$. Therefore, we can approximate the probability of $P((X \cap Y)|A)$ by substituting $C_f(B, C)$ for $C_f((X, Y)|A)$ in Eq. (5).

$$P_a((X \cap Y)|A) = P(X|A) \times P(Y|A) \times C_f(B, C) \tag{6}$$

where $P_a$ is the approximate probability. Similarly, if we have the probability of a itemset, we can easily determine its support count with the following equation.

$$|V| = P(V) \times |U| \quad P(V) = \frac{|V|}{|U|} \tag{7}$$

where $|V|$ is the total number of transactions in which the itemset $V$ is present, and $|U|$ represents the total number of transactions in the dataset. Similarly, if $U$ is the set of all transactions in the dataset, $A$, $B$, $C$, $X$, and $Y$ are the itemsets representing the set of

transactions that are the subsets of $U$. This means that $|V|$ is the support count in the context of itemset mining. Like the ProbBF algorithm, the proposed algorithm ProbDF also uses the following two equations for predicting the support count of $k$-$size$ itemsets for $k \geq 3$.

$$|A \cap B \cap C| = \frac{|A \cap B| \times |A \cap C| \times |B \cap C| \times |U|}{|A| \times |B| \times |C|} \tag{8}$$

$$|Pref \cap A \cap B \cap C| = \frac{|Pref \cap A \cap B| \times |Pref \cap A \cap C| \times |Pref \cap B \cap C| \times |Pref|}{|Pref \cap A| \times |Pref \cap B| \times |Pref \cap C|}. \tag{9}$$

These two equations are reduced forms of Eq. (6). When $k = 3$, Eq. (8) is used, while Eq. (9) is used for $k \geq 4$. In this equation, the last 3 items of a $k -$ itemset are represented with $A$, $B$, and $C$ respectively. $Pref$ (for prefix) is an itemset of size $k - 3$. If $HIJKL$ is a 5-$itemset$, for example, then $J$, $K$, and $L$ are the $A$, $B$, and $C$ respectively, and $HI$ is the $Pref$ itemset. The importance of the recursive application of Eq. (6) in the form of Eq. (9) is that the sample space $|pref|$ is continuously reducing. It is also important to note that support counts of all the sets used in Eq. (9) would already be known.

## Enforcing limits on predictions

If $X = (A \cup B)$, $Y = (A \cup C)$ and $Z = (B \cup C)$, according to the anti-monotone property of the support finding process, the ProbDF introduces the upper bound ($Ub$) on support prediction as follows.

$$\sigma(A \cup B \cup C) \leq min(X, \ Y, \ Z) = Ub. \tag{10}$$

Similarly, a lower bound on the support of any 3-itemset could be defined using the following equation.

$$\sigma(\{a, b, c\}) \geq \sigma(\{a, b\}) + \sigma(\{a, c\}) - \sigma(\{a\}) \tag{11}$$

This equation was first used in Max-miner (*Bayardo, 1998*). Consequently, a lower bound ($Lb$) on the support prediction of a 3-itemset could be defined using the following equation.

$$\sigma(A \cup B \cup C) \geq max \begin{pmatrix} \sigma(X) + \sigma(Y) - \sigma(A), \\ \sigma(X) + \sigma(Z) - \sigma(B), \\ \sigma(Y) + \sigma(Z) - \sigma(C) \end{pmatrix} = Lb. \tag{12}$$

The ProbDF clipps the predicted support on any of these theoratical bounds whenever crossed. These bounds also help to establish a limit on the maximum error. Let $\vartheta$ represents this maximum error given as follows.

$$Error \leq \vartheta = Ub - Lb - 1. \tag{13}$$

In the above equation bounds the error $[0, \vartheta])$. The value $-1$ in the above equation represents a unique situation, which is when $\sigma(A \cup B \cup C) = min(X, \ Y, \ Z)$. At this point

$Lb = Ub$. It has been shown in *Sadeequllah et al. (2024)* that these bounds improves the predictions of the ProbDF greatly.

## THE PROPOSED ALGORITHM, PROBDP

In this section, we present the depth-first based proposed algorithm ProbDF. This algorithm generates frequent itemsets based on the set-enumeration tree (*Rymon, 1992*) model. Every candidate, represented by a node N, has two items of information: the item itself, and a vector of supports of its all possible children. Every node represents a $k$-size frequent itemset, where $k-1$ items are stored in a separate vector *itemset* and only the $k^{th}$ item is stored in the node. The size of the *itemset* vector is equal to the total number of frequent items. The item at the root node is represented by Ø and the support is set to the total number of transactions. This is because ProbDF treats the root node as the universal set $U$. To generate all frequent itemsets and print them to secondary storage, the ProbDF uses the following three algorithms.

This algorithm has two for loops, the outer and the inner. The outer loop creates a node for each frequent child in the support vector of the current node. The inner loop constructs the support vector for each node created. This algorithm calculates the support count of the new itemsets probabilistically using Eq. (9). This support is trimmed at either boundary of $[Lb, \ Ub]$ if it goes beyond this interval. These boundaries, $Lb$ and $Ub$, are given by Eqs. (12) and (10) respectively.

### Introducing promotion pruning in ProbDF

The parent equivalence pruning (PEP) feature, first introduced in *Bayardo (1998)* and elaborated in detail in *Sadeequllah et al. (2024)*, has been tuned to the needs of frequent item mining in *Aryabarzan, Minaei-Bidgoli & Teshnehlab, (2018)*, *Deng, (2016)* under the name promotion method. According to this method if $F$ is a $k$-size frequent itemset and $S$ is a set of items representing the frequent 1-item extensions of $F$ such that $\sigma(F) = \sigma(F \cup i) \ \forall i \in S$, then $\sigma(F) = \sigma(F \cup j) \forall j \in \mathcal{P}(\mathcal{S})$, where $\mathcal{P}(\mathcal{S})$ is the power set of S. If $M = \{F \cup j | j \in \mathcal{P}(S) \wedge size(j) > 1\}$, it is important to note that $M$ is the set of frequent itemsets that are directly generated (*i.e.*, bypassing the candidate generation and test strategy of the FIM algorithms). The experiments show the promotion method significantly improves the time and space complexity of the FIM algorithms (*Aryabarzan, Minaei-Bidgoli & Teshnehlab, 2018*; *Deng, 2016*). It prunes complete branches of the search space rooted at promoted items (the frequent items in set $S$ are known as promoted items). The Algorithms 4 and 5 are modified versions of the Algorithms 3 and 4, respectively that incorporate the promotion method in ProbDF algorithm. The changes in these algorithms from their previous versions are highlighted in bold-face font. The promoted items in these algorithms are stored in a new vector $p\_items$ in every node.

**Example:** The ProbDF algorithm has been applied to the dataset shown in Table 2. This dataset is passed as an argument to the main algorithm, probDFAlgo (Algorithm 2), along with the support threshold of two transactions. For this support threshold, all items present in the dataset are already frequent. The probDFAlgo first calls the

**Input**: $\mathcal{D}$: Transactions Database, $min\_count$: Support Threshold
**Output**: $F1$ : A vector of frequent items along with their support arranged in support ascending order

1. $F1 = \emptyset$ //As <item, support>*
2. $foreach\ tran_i \in \mathcal{D}, \forall i = 1, 2, \ldots, |\mathcal{D}|\ do:$
3. | $for\ item \in t\ do:$
4. | | $if\ NOT\ F1.isItemPresent(item)$
5. | | | $F1.addItem(item, 0)$ //initialize *item* with 0 support
6. | | $F1.incItemSupport(item)$ //increment item support by 1
7. $foreach\ f_i \in F1, \forall i = 1, 2, \ldots, |F1|\ do:$
8. | $if\ f_i.support < min\_count$
9. | | $F1.deleteItem(f_i)$ //delete infrequent item
10. $x_i.support \le x_{i+1}.support, \forall(x_i, x_{i+1}) \in F1$ //sorting F1
11. $Return\ F1$

(a)

**Input**: $\mathcal{D}$: Transactions Database, min_sup: Support Threshold
**Output**: A file with all discovered frequent itemsets

1. $min\_count = \lceil min\_sup \times |\mathcal{D}| \rceil$ //min support count
2. $F1 = getFrequentItems(D, min\_count)$
3. $nodsList = \emptyset$ //List of nodes initialized to null list
4. $sib\_sups = CREATE\ vectore(size = |F1|)$
5. $for\ i = 1, 2, \ldots, |F1|$
6. | $sib\_sups[i] = F1.get(i)$ //from F1, retrieve ith item's support
7. | $nodsList[i] = CREATE\ node < item, child\_sups *>$
8. | $nodsList[i].item = i$
9. | $nodsList[i].child\_sups = CREATE\ vector(size = |F1| - i)$
10. $foreach\ tran_i \in \mathcal{D}, \forall i = 1, 2, \ldots, |\mathcal{D}|\ do:$
11. | $tranItems = \emptyset$ //stores transaction items indices in F1
12. | $foreach\ item \in tran_i\ do:$
13. | | $if\ F1.isItemPresent(item)$
14. | | | $tranItems.add(F1.getIndex(item))$
15. | $Sort(tranItems)$ //sort indices in ascending order
16. | $for\ i = 1, 2, \ldots, |tranItems| - 1$
17. | | $for\ j = i + 1, i + 2, \ldots, |tranItems|$
18. | | | $childInd = tranItems[j] - tranItems[i] - 1$
19. | | | $nodsList[tranItems[i]].child\_sups[childInd]++$
20. $delete\ \mathcal{D}$ //transaction database no more required
21. $itemset = CREATE\ vector(size = |F1|)$
22. $foreach\ nd_i \in nodsList, \forall i = 1, 2, \ldots, |nodsList|\ do:$
23. | $genFItems(|D|, itemset, 0, nd_i, |F1|, min\_count)$

(b)

**Input**: pref_sup: support of itemset $pref$, itemset:, the itemset $pref$, is_size: size of itemset $pref$, a_nod: node for itemset $Pref \cup A$, sib_sups: vector of supports of itemset $Pref \cup A$ and its siblings, fi_count: total frequent items, min_count: minimum support count
**Output**: Write frequent itemsets to file

1. $is\_size = is\_size + 1$
2. $itemset[is\_size] = a\_node.item$
3. $s\_offset = fi\_count - |sib\_sups|$
4. $A = [a\_node.item - s\_offset]$ //support of itemset $Pref \cup A$
5. $printToFile(A, is\_size, a\_node)$ // print to file itemset $Pref \cup A$
6. $if\ a\_node.child\_sups = \emptyset$
7. | $return$ //return if a_node is the leaf node
8. $child\_sups = a\_node.child\_sups$
9. $offset = fi\_count - |child\_sups|$
10. $nodsList = \emptyset$
11. $outer: for\ i = 1, 2, 3, \ldots, |chil\_sups|$
12. | $X = child\_sups[i]$ //X = Pref ∪ A ∪ B
13. | $if\ X < min\_count$
14. | | $go\ to\ outer$
15. | $Repeat\ unitill\ a\_node.item = offset + i$
16. | | $a\_node.advance$
17. | $zsups = a\_node.child\_sups$
18. | $if |zsups| > 0$
19. | | $z\_offset = fi\_count - |zsups|$
20. | | $inner: for\ j = z\_offset - offset, z\_offset - offset + 1, \ldots, |child\_sups|$
21. | | | $Y = child\_sups[j]$ //Y = Pref ∪ A ∪ C
22. | | | $if\ Y < min\_count$
23. | | | | $go\ to\ inner$
24. | | | $Z = zsups[offset + j - z\_offset]$ //Z = Pref ∪ B ∪ C
25. | | | $if\ Z < min\_count$
26. | | | | $go\ to\ inner$
27. | | | $B = sib\_sups[offset + i - soffset]$
28. | | | $C = sib\_sups[offset + j - soffset]$
29. | | | $supp = \frac{X \times Y \times Z \times Pref\_sup}{A \times B \times C}$
30. | | | $if\ supp > min(X, Y, Z)$
31. | | | | $supp = min(X, Y, Z)$
32. | | | $if\ supp < max(X + Y - A, X + Z - B, Y + Z - C)$
33. | | | | $supp = max(X + Y - A, X + Z - B, Y + Z - C)$
34. | | | $if\ supp \ge min\_count$
35. | | | | $if\ first$
36. | | | | | $new\_sups = CREATE\ vector(size = freq\_count - offset - j)$
37. | | | | | $first = false$
38. | | | | | $new\_sups.add(supp)$
39. | | | | $else\ if\ NOT\ first$
40. | | | | | $new\_sups.add(0)$ //0 support is entered for infrequent child
41. | | $new\_nod = CREATE\ node < item, child\_sups *>$
42. | | $new\_nod.item = offset + i$
43. | | $if\ first$
44. | | | $new\_nod.child\_sups = \emptyset$
45. | | $else$
46. | | | $new\_nod.child\_sups = new\_sups$
47. | | $nodsList.addNode(new\_next)$
48. $foreach\ nd_i \in nodsList, \forall i = 1, 2, \ldots, |nodsList|\ do:$
49. | $genFItems(A, itemset, is\_size, nd_i, fi\_count, min\_count)$

(c)

**Figure 1 ProbDF algorithm.** (A) genFrequentItems routine (Algorithm 1) (B) the main algorithm ProbDF routing (Algorithm 2) (C) genFItemsets routine (Algorithm 3).

### Algorithm 1

This algorithm, presented in Fig. 1A, receives the transaction dataset $\mathcal{D}$ as input. This dataset has a total of $|\mathcal{D}|$ transactions, and every transaction $t \subseteq I$. This algorithm reads every transaction $t \in \mathcal{D}$ and every item $i \in t$, and builds $F1$. The $F1$ is a collection of items, present in $\mathcal{D}$, along with support of these items. If the support of an item is less than the user-supplied support threshold *min_sup*, it is deleted. That is, $F1$ is left with frequent items only. It is worth noting that frequent items in $F1$ are sorted in support ascending order. This algorithm is called by Algorithm 2, which is the main algorithm the ProbDF.

### Algorithm 2

In Fig. 1B , the main algorithm ProbDF is presented. In order to create the list of frequent items (*i.e.*, the $F1$), this algorithm first calls Algorithm 1. Though the root node is not explicitly declared, it creates a vector to store support values of the frequent items as children of the implicit root node. Every node stores the support of its children in such a vector and we will call it support vector from this point onward. It is important to note that, in probDF algorithm, the frequent items are not directly stored in memory, rather implicitly represented by the indices of these vectors. This scheme is adopted in order to make ProbDF time and space efficient. From line 5 to line 9, it creates a node for every frequent item in $F1$, assign it the item index, and create a vector to store support values for all its possible children. From line 10 to line 15, it retrieves each transaction from $\mathcal{D}$, deletes infrequent items from this transaction, retrieves indices of the frequent items left, and sort these indices in ascending order. From line 16 to line 19, these indices are used to increment the support values of appropriate 2-itemsets. It also creates a vector *itemset*. This vector stores every frequent $k$-itemset that is generated by the algorithm and written to the disk. Lastly, it calls Algorithm 3 and passes it the nodes, one by one, that it has created for the frequent items in $F1$. Algorithm 3 writes these nodes to disk and generates recursive calls to itself to create further children of these nodes.

### Algorithm 3

This algorithm, presented in Fig. 1C, has a major share in the ProbDF algorithm. This algorithm first prints the node passed as parameter. If the node represents $k$-itemset, it loads the node label as $k$th item on *itemset* vector (the $k$-1 items are already loaded on *itemset*). It also retrieves support of the node from the support vector of its parent node and writes the itemset the node represents to disk along with its support. It is important to note that the item whose support is stored at first location in the support vector, is named as offset (also *s_offset* and *z_offset*). For a node N, the *offset* $\geq N.item + 1$. The *offset* $= N.item + 1$ only if the item $N.item + 1$ is frequent, otherwise *offset* $> N.item + 1$. This is because storing supports of the starting items, which are infrequent, is wastage of time and memory.

### Algorithm 4

This algorithm, shown in Fig. 2A , is a modified version of the Algorithm 2 that incorporates new instructions, written in bold face, to introduce the promotion pruning concept in ProbDF. Every node has now an extra *p_items* vector to store promoted items. It creates a global stack data structure to temporarily store promoted items of a node. From line 22 to line 28, it checks the children of every frequent child node of the root node for promotion. If there are one or more children found as promoted items, they are pushed on the stack. The support count value of the promoted item is incremented by the total transactions count $|\mathcal{D}|$. The purpose of this incrementation is to differentiate the support count of promoted itemset from the other frequent itemsets. This point is further explained in the next algorithm, the Algorithm 5. From line 29 to 31, the *p_items* vector of the size of stack is created and the stack is emptied into it.

getFrequentItems routine and receives in return the list of frequent items, arranged in ascending support order, as $F1$. For the data in Table 2, the $F1$ is $\{S, T, W, X, Y, Z\}$.

The probDFAlgo starts traversing the search space from line 5 to line 9, creating the child nodes of the implicit root node for every frequent item in $F1$ as a frequent child. This is shown in Fig. 3A. The support of the implicit root node is set at $|\mathcal{D}|$ because we use the diffset support computations method (*Zaki & Gouda, 2003*). For the data in Table 2, $|\mathcal{D}|$ is 10. Every child node also has a vector to store support values of all its possible children, and the probDFAlgo determines these values from the transactions database $\mathcal{D}$ from lines 10–19. Lines 22 and 23 perform further exploration of the nodes in Fig. 3A. The probDFAlgo calls the genFItems routine for this purpose.

**Input**: $\mathcal{D}$: Transactions Database, $min\_sup$: Support Threshold
**Output**: A file with all discovered frequent itemsets

1. $min\_count = \lceil min\_sup \times |\mathcal{D}| \rceil$ //min support count
2. $F1 = getFrequentItems(\mathcal{D}, min\_count)$
3. $nodsList = \emptyset$ //List of nodes initialized to null list
4. $sib\_sups = CREATE\ vectore(size = |F1|)$
5. $for\ i = 1, 2, \ldots, |F1|$
6. | $sib\_sups[i] = F1.get(i)$ //from F1, retrieve ith item's support
7. | $nodsList[i] = CREATE\ node < item, child\_sups *, p\_items *>$
8. | $nodsList[i].item = i$
9. | $nodsList[i].child\_sups = CREATE\ vector(size = |F1| - i)$
10. $foreach\ tran_i \in \mathcal{D}, \forall i = 1, 2, \ldots, |\mathcal{D}|\ do:$
11. | $tranItems = \emptyset$ //stores transaction items indices in F1
12. | $foreach\ item \in tran_i\ do:$
13. | | $if\ F1.isItemPresent(item)$
14. | | | $tranItems.add(F1.getIndex(item))$
15. | $Sort(tranItems)$ //sort indices in ascending order
16. | $for\ i = 1, 2, \ldots, |tranItems| - 1$
17. | | $for\ j = i + 1, i + 2, \ldots, |tranItems|$
18. | | | $childInd = tranItems[j] - tranItems[i] - 1$
19. | | | $nodsList[tranItems[i]].child\_sups[childInd]++$
20. $delete\ \mathcal{D}$ //transaction database no more required
21. $\boldsymbol{GLOBAL\ Stack = CREATE\ vector(size = |F1|)}$
22. $for\ i = 1, 2, \ldots, |F1|$
23. | $top = 0$
24. | $for\ j = 1, 2, \ldots, |nodsList[i].child\_sups|$
25. | | $if\ nodsList[i].child\_sups[j] = sib\_sups[i]$
26. | | | $top = top + 1$
27. | | | $stack[top] = i + j + 1$ //frequent item index
28. | | | $nodsList[i].child\_sups[j] = nodsList[i].child\_sups[j] + |\mathcal{D}|$
29. | $nodsList[i].p\_items = CREATE\ vector(size = top)$
30. | $for\ k = 1, 2, \ldots, top$
31. | | $nodsList[i].p\_items[k] = stack[k]$
32. $itemset = CREATE\ vector(size = |F1|)$
33. $foreach\ nd_i \in nodsList, \forall i = 1, 2, \ldots, |nodsList|\ do:$
34. | $genFItems(|\mathcal{D}|, itemset, 0, nd_i, |F1|, min\_count, |\mathcal{D}|)$

(a)

**Input**: $pref\_sup$: support of itemset $pref$, itemset:, the itemset $pref$, $is\_size$: size of itemset $pref$, $a\_nod$: node for itemset $Pref \cup A$, $sib\_sups$: vector of supports of itemset $Pref \cup A$ and its siblings, $fi\_count$: total frequent items, $min\_count$: minimum support count, trans: total transactions

**Output**: Write frequent itemsets to file

1. $is\_size = is\_size + 1$
2. $itemset[is\_size] = a\_node.item$
3. $s\_offset = fi\_count - |sib\_sups|$
4. $A = [a\_node.item - s\_offset]$ //support of itemset $Pref \cup A$
5. $printToFile(A, is\_size, a\_node)$ // print to file itemset $Pref \cup A$
6. $if\ a\_node.child\_sups = \emptyset$
7. | $return$ //return if $a\_node$ is the leaf node
8. $\boldsymbol{parent\_nod = a\_node}$
9. $child\_sups = a\_node.child\_sups$
10. $offset = fi\_count - |child\_sups|$
11. $nodsList = \emptyset$
12. $outer: for\ i = 1, 2, 3, \ldots, |chil\_sups|$
13. | $X = child\_sups[i]$ //$X = Pref \cup A \cup B$
14. | $if\ X < min\_count \vee X > trans$
15. | | $go\ to\ outer$
16. | | $\boldsymbol{top = 0}$
17. | $Repeat\ unitill\ a\_node.item = offset + i$
18. | | $a\_node.advance$
19. | $zsups = a\_node.child\_sups$
20. | $if |zsups| > 0$
21. | | $z\_offset = fi\_count - |zsups|$
22. | | $inner: for\ j = z\_offset - offset, z\_offset - offset + 1, \ldots, |child\_sups|$
23. | | | $Y = child\_sups[j]$ //$Y = Pref \cup A \cup C$
24. | | | $if\ Y < min\_count \vee X > trans$
25. | | | | $go\ to\ inner$
26. | | | $Z = zsups[offset + j - z\_offset]$ //$Z = Pref \cup B \cup C$
27. | | | $if\ Z < min\_count$
28. | | | | $go\ to\ inner$
29. | | | $\boldsymbol{if\ Z > trans}$
30. | | | | $\boldsymbol{Z = Z - trans}$
31. | | | $B = sib\_sups[offset + i - soffset]$
32. | | | $C = sib\_sups[offset + j - soffset]$
33. | | | $supp = \frac{X \times Y \times Z \times Pref\_sup}{A \times B \times C}$
34. | | | $if\ supp > min(X, Y, Z)$
35. | | | | $supp = min(X, Y, Z)$
36. | | | $if\ supp < max(X + Y - A, X + Z - B, Y + Z - C)$
37. | | | | $supp = max(X + Y - A, X + Z - B, Y + Z - C)$
38. | | | $if\ supp \geq min\_count$
39. | | | | $if\ first$
40. | | | | | $new\_sups = CREATE\ vector(size = freq\_count - offset - j)$
41. | | | | | $first = false$
42. | | | | $\boldsymbol{if\ supp = X}$
43. | | | | | $top = top + 1$
44. | | | | | $stack[top] = offset + j$
45. | | | | | $new\_sups.add(supp + trans)$
46. | | | | $else$
47. | | | | | $new\_sups.add(supp)$
48. | | | $else\ if\ NOT\ first$
49. | | | | $new\_sups.add(0)$ //0 support is entered for infrequent child
50. | $new\_nod = CREATE\ node < item, child\_sups *>$
51. | $new\_nod.item = offset + i$
52. | $if\ first$
53. | | $new\_nod.child\_sups = \emptyset$
54. | $else$
55. | | $new\_nod.child\_sups = new\_sups$
56. | $if\ top = 0$
57. | | $new\_nod.p\_items = parent\_nod.p\_items$
58. | $else$
59. | | $newp\_items = CREATE\ vector(size = top + |parent\_nod.p\_items|)$
60. | | $newp\_items.addAll(parent_{nod}.p_{items}, 0, |parent_{nod}.p_{items}|)$
61. | | $newp\_items.addAll(stack, 0, top)$
62. | | $new\_next.p\_items = newp\_items$
63. | $nodsList.addNode(new\_next)$
64. $foreach\ nd_i \in nodsList, \forall i = 1, 2, \ldots, |nodsList|\ do:$
65. | $genFItems(A, itemset, is\_size, nd_i, fi\_count, min\_count, trans)$

(b)

**Figure 2 ProbDF algorithm with promotion pruning.** (A) Algorithm 2 Extended (Algorithm 4) (B) Algorithm 3 Extended (Algorithm 5).

**Algorithm 5**

This algorithm, presented in Fig. 2B, is modified version of the algorithm 3. This algorithm also incorporates new instructions to bring the concept of promotion into the ProbDF algorithm. These new instructions (or part of an instruction) are distinguished as boldface text. At line 8, the reference of the current frequent node $a\_node$ is stored in a new reference variable (pointer) because $a\_node$ keeps advancing down the algorithm in tracking the support of $Z$. The child item $X$ or $Y$ is not selected for candidate generation if its support is greater than $|\mathcal{D}|$. This is because greater support than $|\mathcal{D}|$ means the item is promoted and it is already stored in $p\_items$. This is according to plan as we discussed in the beginning of this section that the concept of promotion prunes complete branches of the search space rooted at promoted items. The purpose of storing the support of promoted item is only to determine the support of $Z$. From line 42 to 45, if the new frequent candidate is a promoted item, it is pushed on the stack and its support is stored in the newly created support vector incremented by $|\mathcal{D}|$. From 56 to 62, the $p\_items$ vector of the new node is decided. If there is no new promoted item, $i.$ $e.$, the stack is underflow, the parent vector of $p\_items$ is passed as it is to the new node. Otherwise, a new vector is created to store the promoted items of both the parent node as well as that of the new node.

The genFItems recursively explores the nodes it receives as arguments in depth-first order. It creates a child node for frequent children only. The support of the children of the top-level nodes (*i.e.*, nodes listed in Fig. 3A) was calculated from the transaction database by the calling routine probDFAlgo. Therefore, the top-level nodes and their children nodes are shown as round nodes to indicate their support is calculated from the transaction database. These nodes are shown in Fig. 3B. The later nodes down the tree, starting from level-3, are shown as rectangular nodes to indicate their support is approximated using PSPM. It is worth noting that the proposed algorithm creates children nodes of a parent node in breadth-first order, but traverses these children nodes in depth-first order. This is also evident from Figs. 3A and 3B. All the children nodes of the root node are created in breadth-first order, as shown in Fig. 3A. Similarly, all the children nodes of the node $S$ are also created in breadth-first order in Fig. 3B.

The rectangular nodes start to generate when the genFItems call itself recursively for the first time. This time, the nodes created have their support approximated in the previous call to genFItems, which was made from probDFAlgo. However, the rectangular children nodes of the round node $SW$, shown in Fig. 3C, are all infrequent, and the first recursive call to genFItems returns without making any addition to the search space tree.

The second recursive call to the genFItems routine in Fig. 3D is made for the round node $SX$. This call creates two rectangular children, $SXY$ and $SXZ$. Both of these children are frequent. Figure 3E shows that the third recursive call creates the rectangular node $SXYZ$. The diagram of the complete search space is shown in Fig. 3F.

The support calculated using the PSPM is probabilistic; therefore, these values are continuous. For example, to calculate the probabilistic support for the itemset $\{S, W, Y\}$ using Eq. (8), the following calculations are used:

$$support(SWY) = \frac{|S \cap W| \times |S \cap Y| \times |W \cap Y| \times |U|}{|S| \times |W| \times |Y|}$$

$$support(FDB) = \frac{2 \times 3 \times 4 \times 10}{4 \times 6 \times 8} = 1.25.$$

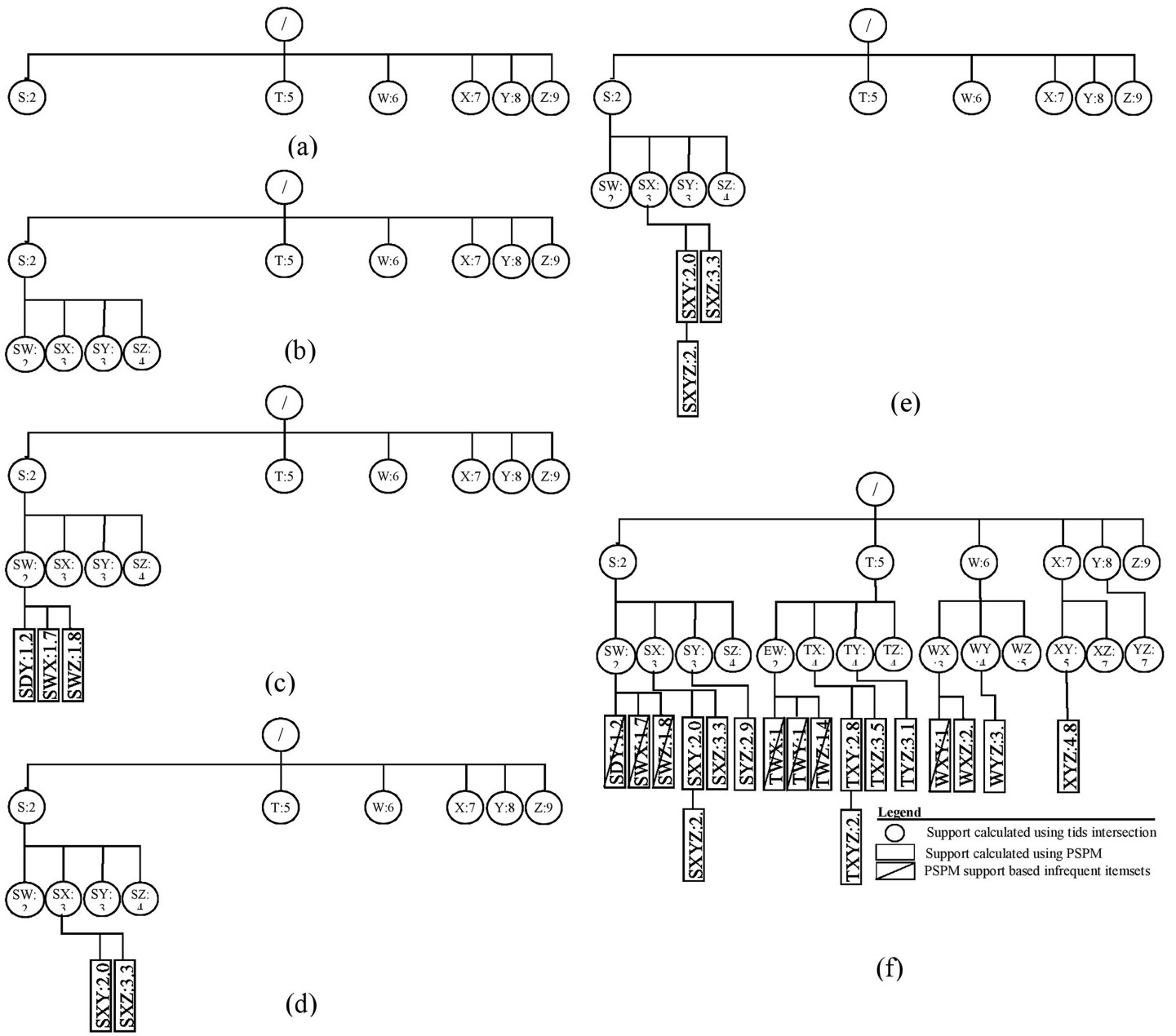

**Figure 3 ProbDF applied to data in Table 1.**

Similarly, to calculate the support of *SXYZ*, put *Pref = S, A = X, B = Y* and *C = Z* in Eq. (9), and the answer is 2.17 (the more precise answer is from ProbDF).

$$|S \cap X \cap Y \cap Z| = \frac{|S \cap X \cap Y| \times |S \cap X \cap Z| \times |S \cap Y \cap Z| \times |S|}{|S \cap X| \times |S \cap Y| \times |S \cap Z|}$$

$$support(SXYZ) = \frac{2.01 \ \times 3.33 \times 2.91 \times 4}{3 \times 3 \times 4} = 2.164 \approx 2.17.$$

**Complexity analysis of the ProbDF**

The total runtime of the ProbDF algorithm comprises two distinct parts: the time required for the support calculation of frequent itemsets of size 1 and 2, and the time required to generate and evaluate candidate itemsets belong to the level 3 and beyond. Let $n$ represent the total number of frequent items and $T$ represent the total number of transactions in a given dataset. In the worst-case scenario, the support calculation for frequent itemsets of size 1 and 2 assumes each frequent item is present in every transaction, resulting in a time complexity of $O(n \times T)$. Generating all possible candidate itemsets is a combinatorial problem that involves generating $2^n$ subsets of the set of $n$ frequent items, leading to a time complexity of $O(2^n)$. Therefore, the total runtime of the ProbDF algorithm in the worst case is $O(n \times T) + O(2^n)$. When considering the average runtime complexity, replacing $n$ with the Average Transaction Length ($ATL$), we obtain $O(ATL \times T) + O(2^n)$.

The space complexity of ProbDF can be determined by examining the example presented in Fig. 3. The space required for the vectors, storing child itemsets support, at level 1 is $\frac{(n)(n-1)}{2}$, at level 2 is $\frac{(n-1)(n-2)}{2}$, and so on. For the second-to-last level, it is 1, and for the last level, it is 0. Let $\varphi$ represent the height of the frequent itemset tree (*i.e.*, the length of the longest frequent itemset). Then the space complexity is $\varphi \left[ \frac{(n)(n-1)}{2} + \frac{(n-1)(n-2)}{2} + \ldots + 1 + 0 \right] + \varphi$, which can be simplified to $O(\varphi n^2 + \varphi)$.

## EXPERIMENTAL RESULTS

In this section, we present a comprehensive evaluation of the performance of the proposed algorithm ProbDF. For this purpose, ProbDF has been implemented in Java version 8. The operating system used was Windows 10, and the hardware platform was a 10th generation Intel Core i5. The reason for selecting JAVA are the problems, regarding implementation of the FIM algorithms, identified in *Fournier-Viger et al. (2017)*, *Luna, Fournier-Viger & Ventura (2019)*. The author believes that the original versions of different FIM algorithms have been implemented in different languages, and across diverse hardware platforms, which greatly affect their performance. The worst is that some algorithms have no implementation available whatsoever. To deal with these problems, the authors have proposed an online Java based open source FIM library, called SPMF and accessible at https://www.philippe-fournier-viger.com/spmf/v2/index.php. For a fear comparison, the algorithms should not only have the same hardware and software platform, but should also be implemented and tested in the same language and the same compiler. Due to this, four state-of-the-art FIM algorithms–namely FP Growth (*Grahne & Zhu, 2005*), PrePost+ (*Deng & Lv, 2015*), dFIN (*Deng, 2016*) and negFIN (*Aryabarzan, Minaei-Bidgoli & Teshnehlab, 2018*), have been selected from the SPMF library. However, all these

algorithms are the exact algorithms, *i.e.*, they generate no false positive or false negative. For a fair comparison, we have also selected three recently published approximate FIM algorithms–namely ProbBF (*Sadeequllah et al., 2024*), HashEclat (*Zhang–et al., 2019*), and RSB-PFI (*Valiullin et al., 2021*). All these algorithms have been recoded in JAVA. While the original version of ProbBF is implemented in C++ and uses a set enumeration tree to store the discovered frequent items, the Java version uses a simple queue data structure to conserve memory more efficiently. It is due to the fact that some FIM algorithms store the discovered frequent itemsets in memory, while others write them to file. For uniform comparison of bench mark algorithms in *Sadeequllah et al. (2024)*, all the algorithms were forced to store all of the discovered frequent itemsets in memory resident data structures. However, to commpare only the frequent itemset generation capacity of these algorithms, we force these algorithms to store these frequent itemsets neither in memory and nor write them files, *i.e.*, the discovered frequent itemsets are only counted and then discarded. Moreover, the ProbDF will also be made available in the future distributions of SPMF liberary.

To benchmark the performance of the ProbDF on dense data, we have selected five real-world datasets. These datasets are the standard benchmarking datasets frequently present in the frequent itemset mining literature. These datasets are Chess, Connect, Accidents, Pumsb, and Pumsb_star, which are available for free downloads from the FIMI repository (http://fimi.ua.ac.be). The input parameters for all eight algorithms included in this comparison, along with their corresponding outcomes, are presented in Table 3. Additionally, Table 4 presents a description of the various symbols and notations used throughout this article.

In this section, we first evaluate the time and memory efficiency, respectively, of the ProbDF algorithm compared to the seven benchmark algorithms selected for comparison. Subsequently, analysis of our algorithm's capability to generate all potential frequent itemsets is presented.

## Time efficiency of ProbDF

In Fig. 4A , the total runtime of the seven benchmark algorithms: NegFIN, dFIN, PrePost+, FP-Growth, HashEclat, RSB-PFI, and ProbBF, has been compared against the run time of the ProbDF algorithm. In this comparison, these algorithms have been applied to five benchmark datasets: Chess, Connect, Accidents, Pumsb, and Pumsb_star, and the results are shown in Fig. 4A.

As previously mentioned, the most processing-intensive operation is support discovery in frequent itemset mining. This is because it requires to examine all the relevant transactional data to be able to determine support of an itemset. Since the ProbDF does not use any transactional data in the support finding operation, this fact makes it the most efficient algorithm. This efficiency is visible in Fig. 4A  across all the benchmark datasets and against all seven state-of-the-art algorithms. As stated earlier, all these algorithms in this comparison are configured in such a way that they discard frequent itemsets soon after they are generated. That is, the frequent itemsets are neither stored in memory nor written to disk. The reason for this setting is to strictly restrict the comparison to the very basic

**Table 3 Analysis of the input parameters of the algorithms selected for comparison and their runtime values.**

**Support values (%)**

| Algorithm | 50% | 60% | 70% | 80% | Dataset |
|-----------|-----|-----|-----|-----|---------|
| NegFIN | 0.146 | 0.054 | 0.029 | 0.024 | Chess |
| dFIN | 0.184 | 0.073 | 0.043 | 0.036 | |
| PrePost+ | 0.246 | 0.097 | 0.044 | 0.036 | |
| FPGrowth | 0.463 | 0.168 | 0.067 | 0.028 | |
| HashEclat | 0.38 | 0.145 | 0.063 | 0.027 | |
| RSB-PFI | 0.52 | 0.185 | 0.081 | 0.04 | |
| ProbBF | 0.12 | 0.044 | 0.025 | 0.012 | |
| ProbDF | 0.048 | 0.023 | 0.015 | 0.012 | |
| | **50%** | **60%** | **70%** | **80%** | |
| NegFIN | 0.626 | 0.601 | 0.546 | 0.527 | Connect |
| dFIN | 0.993 | 0.715 | 0.576 | 0.546 | |
| PrePost+ | 5.381 | 1.402 | 0.831 | 0.625 | |
| FPGrowth | 10.865 | 3.064 | 1.645 | 0.787 | |
| HashEclat | 8.83 | 2.537 | 1.256 | 0.663 | |
| RSB-PFI | 13.52 | 4.59 | 1.94 | 0.821 | |
| ProbBF | 0.75 | 0.62 | 0.5 | 0.4 | |
| ProbDF | 0.432 | 0.4 | 0.372 | 0.357 | |
| | **50%** | **60%** | **70%** | **80%** | |
| NegFIN | 27.183 | 3.071 | 1.146 | 0.734 | PUMSB |
| dFIN | 22.668 | 2.85 | 1.09 | 0.709 | |
| PrePost+ | 40.21 | 6.267 | 1.552 | 0.729 | |
| FPGrowth | 92.41 | 11.084 | 2.526 | 0.862 | |
| HashEclat | 159.58 | 19.091 | 4.127 | 1.549 | |
| RSB-PFI | 132.58 | 15.31 | 3.36 | 1.2 | |
| ProbBF | 9.61 | 1.947 | 0.87 | 0.746 | |
| ProbDF | 2.3 | 0.784 | 0.58 | 0.483 | |
| | **40%** | **50%** | **60%** | **70%** | |
| NegFIN | 0.614 | 0.479 | 0.421 | 0.456 | PUMSB_star |
| dFIN | 0.61 | 0.495 | 0.451 | 0.434 | |
| PrePost+ | 0.682 | 0.527 | 0.468 | 0.445 | |
| FPGrowth | 0.877 | 0.575 | 0.475 | 0.453 | |
| HashEclat | 1.27 | 0.7166 | 0.5676 | 0.5564 | |
| RSB-PFI | 1.124 | 0.681 | 0.524 | 0.516 | |
| ProbBF | 0.5148 | 0.4498 | 0.3896 | 0.377 | |
| ProbDF | 0.371 | 0.329 | 0.306 | 0.289 | |
| | **40%** | **45%** | **50%** | **55%** | |
| NegFIN | 3.359 | 2.972 | 2.468 | 2.35 | Accidents |
| dFIN | 2.679 | 2.569 | 2.425 | 2.359 | |
| PrePost+ | 2.68 | 2.567 | 2.387 | 2.367 | |

| Support values (%) | | | | |
|---|---|---|---|---|
| FPGrowth | 3.789 | 3.184 | 2.725 | 2.519 |
| HashEclat | 4.81 | 4.24 | 3.476 | 3.32 |
| RSB-PFI | 5.51 | 4.75 | 3.82 | 3.5 |
| ProbBF | 2.4 | 2.2 | 1.9 | 1.8 |
| ProbDF | 1.502 | 1.47 | 1.436 | 1.417 |

**Table 4 Notations table.**

| Symbol | Description |
|---|---|
| AFIS | Average frequent itemset size |
| $ATL$ | Average transaction length |
| $C_f$ | Correlation factor |
| $\mathcal{D}$ | Transaction dataset |
| dFIN | A FIM algorithm selected for comparison |
| $F1$ | Collection of frequent items with their support counts |
| FIM | Frequent Itemset Mining |
| HashEclat | A FIM algorithm selected for comparison |
| $I$ | An item in the transaction t |
| $K$ | Size of the itemset |
| $Lb$ | Lower bound of support prediction |
| MFI | Maximal Frequent Itemsets |
| $min\_sup$ | Minimum support threshold |
| $N$ | Node in the set-enumeration tree |
| $n$ | Total number of frequent items |
| NegFIN | A FIM algorithm selected for comparison |
| $O$ | Big-O notation for time complexity |
| $P(A)$ | Probability of the event A |
| $\mathcal{P}(X)$ | Power set of set X |
| PEP | Parent equivalence pruning |
| $Pref$ | Prefix itemset |
| PrePost+ | A FIM algorithm selected for comparison |
| ProbBF | Probabilistic breadth-first (algorithm) |
| ProbDF | Probabilistic depth-first (algorithm) |
| PSPM | Probabilistic support prediction model |
| RSB-PFI | A FIM algorithm selected for comparison |
| SPMF | Java based open source FIM algorithms and datasets online liberary at https://www.philippe-fournier-viger.com/spmf/v2/index.php |
| $T$ | A transaction in the dataset $\mathcal{D}$ |
| $T$ | Total number of transactions in the dataset $\mathcal{D}$ |
| $U$ | The universal set representing the set of all transactions in $\mathcal{D}$ |
| $Ub$ | Upper bound of support prediction |
| $\Theta$ | Maximmumm error on predicted (approximated) support |
| $\sigma(X)$ | Support of itemset X |
| $\Phi$ | Height of the frequent itemset tree (*i.e.*, the length of the longest frequent itemset) |

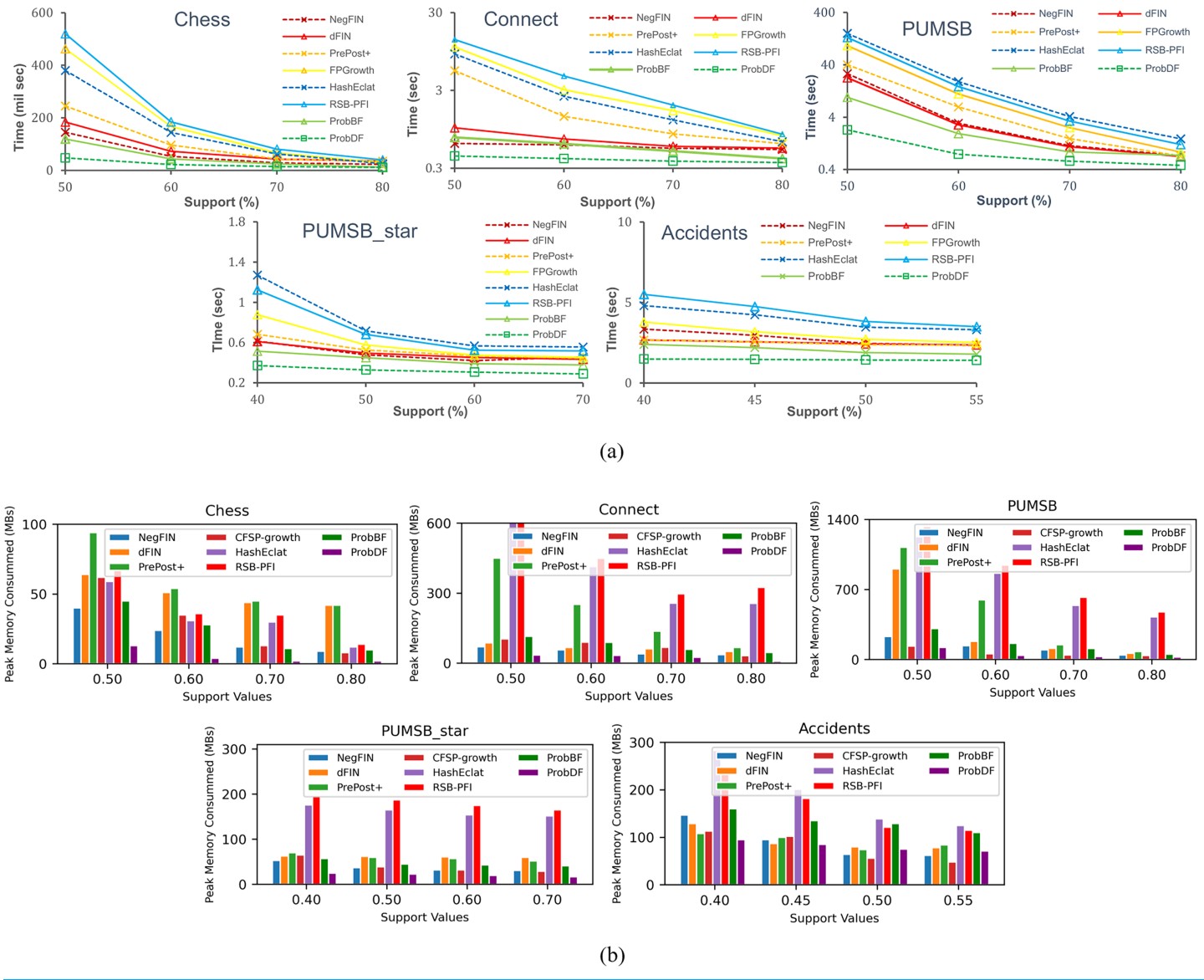

**Figure 4 Evaluating efficiency of the ProbDF algorithm.** (A) Runtime comparison; (B) memory comparison.

memory and computational requirements of the algorithms. We believe that the computational gains of algorithms are diluted by including the writing time (the time it takes to write the frequent itemsets to disk) in the total time the algorithms take to mine the datasets. Similarly, storing frequent itemsets in memory will also thin-out memory savings by the algorithms.

The ProbBF algorithm, as state earlier, has been recoded with queue data structure. However, the unique requirements of Eqs. (8) and (9) enforce that this queue data structure holds three recently traversed tree levels in memory. This contrasts with conventional breadth-first algorithms, which hold only one level in memory. However, the

memory footprint of the Java-coded ProbBF is significantly smaller than that of the original version, as it utilizes a set enumeration tree to store all discovered frequent itemsets. Nevertheless, the Java-coded ProbBF still exhibits slower performance than NegFIN on the Connect dataset, which is notably dense. This discrepancy arises because NegFIN consumes lower memory than ProbBF when it refrains from storing any frequent itemsets in memory. Additionally, ProbBF is further disadvantaged by its lack of utilization of the promotion pruning concept, a feature that NegFIN employs effectively.

The other two approximate algorithms, HashEclat and RSB-PFI, perform poorly on dense datasets. As reported in *Sadeequllah et al. (2024)*, their lack of effective compression of transaction data makes them less suitable options for dense datasets, *i.e.*, these algorithms suit sparse data. On the other hand, ProbDF outperforms all seven algorithms by significant margins due to its simple computations with zero transaction data, efficient depth-first-based search strategy, and effective promotion pruning application.

## Space efficiency of probDF

Since ProbBF and ProbDF algorithms do not calculate support from the transactional data for frequent itemsets of size three or more (*i.e.*, $k \geq 3$), they delete all transaction data after determining all frequent 2-itemsets. This characteristic results in very low memory requirements for these two algorithms. However, ProbBF consumes more memory than ProbDF because its queue data structure stores the three most recently traversed levels of the search space tree. On the other hand, ProbDF's newly proposed efficient depth-first search strategy stores partial results of only a small number of candidate frequent itemsets currently under consideration. Moreover, the application of promotion pruning further reduces the memory requirements of this algorithm by avoiding the storage and exploration of some branches of the search space. In conclusion, the combination of zero transaction data, memory-efficient depth-first search strategy, and promotion pruning renders ProbDF more memory-efficient than all other state-of-the-art FIM algorithms, as shown in Fig. 4B. ProbDF consumes the lowest memory on all benchmark datasets when compared to all competing FIM algorithms.

## Analysis of the real frequent itemsets generation

Since ProbDF is a probabilistic algorithm, the frequent itemsets it generates could be classified as true positive, false negative and false positive. In this subsection, we analyze the strength of the ProbDF algorithm to generate the wanted true positive frequent itemsets, the unwanted false positive frequent itemsets and the missed frequent patterns (*i.e.*, the false negative frequent itemsets). Let us call a frequent itemset as real frequent itemset (*i.e.*, true positive) if its support count is above the min_supp threshold when calculated from the transactional data. Also, call the frequent itemset the ProbDF generates using PSPM be a probabilistic frequent itemset. This analysis has been done on all five selected benchmark datasets. Every dataset has been evaluated for a specific support threshold value, written in parenthesis along the dataset name in the diagrams. For example, in Fig. 5A, the ProbDF has been run on the Chess dataset for a support threshold of 0.4. This figure shows strength of the ProbDF to generate true positive frequent itemsets. For example, the red bar shows

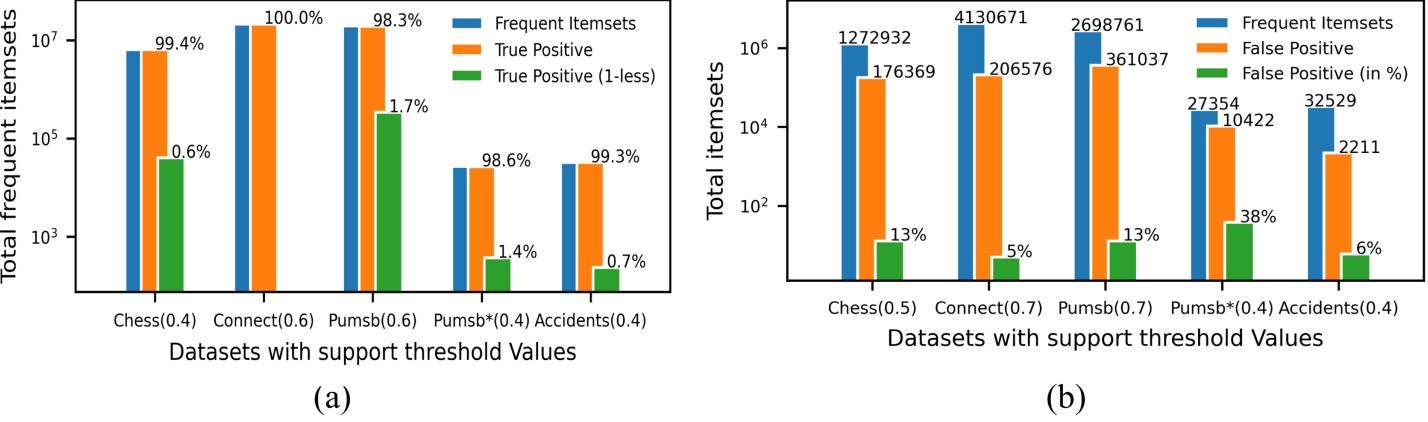

**Figure 5** **Evaluating quality of the output of ProbDF.** (A) True positive analysis. (B) False positive analysis.

the real frequent itemsets for the chess dataset for the support value of 0.4 are 6,438,989. From these, the ProbDF has successfully generated 6,398,631, which are 99.4% of real frequent itemsets. These true positive frequent itesets for the Chess dataset are shown with blue bar in the figure. The remaining real frequent itemsets, which stands at 0.6%, have also been generated by ProbDF, but with one item missing from these itemsets. These itemsets have been shown with the brown bar along with the label 0.6%. The results of the other datasets are not very different than that of the Chess except the Connect. The results for the Connect dataset are unique in that non of the real frequent itemset has been missed by the ProbDF algorithm. This figure clearly shows that the true positive frequent itemsets generated by ProbDF are very close in numbers to the real frequent itemsets. In other words, ProbDF successfully generates majority of the frequent itemsets.

The analysis of the false positive frequent itemsets generated by the ProbDF is presented in Fig. 5B. In this figure, the blue bar represents the total real frequent itemsets for a given dataset. Similarly, the brown bar represents the total false positive frequent itemsets generated by the ProbDF for the given dataset and the mentioned support. These false positive frequent itemset are also represented by the green bar but with different metric, *i.e.*, in percentage. The green bar shows what percent the false positive frequent itemsets are of the real frequent itemsets. This percentage value is written on top of the green bar. The blue bar and the brown bar are also labeled with the number of itemsets they represent. The lowest number of false positive itemsets is recorded for the Connect dataset, while the highest number of false positive itemsets is shown for the Pumsb_star dataset. A reason for this is the density of the database. As mentioned earlier, the Connect dataset is very dense, but the PUMSB_star database is low in density, as this dataset is derived from the PUMSB dataset by deleting all those frequent items whose support is 80% or above. These results clearly indicate that the more the dataset is dense and the support threshold is high, the PSPM produces better results.

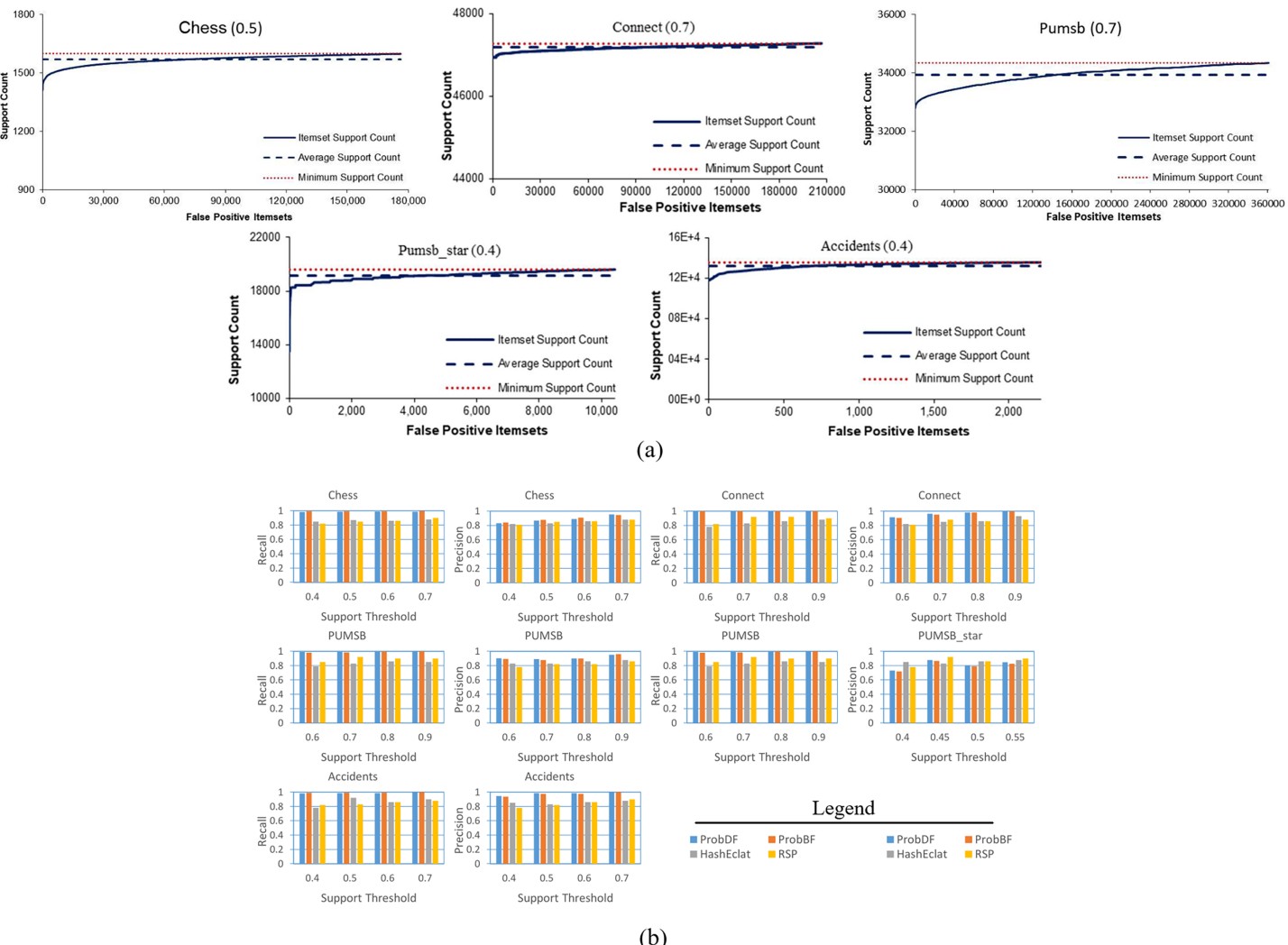

**Figure 6 Further analysis of the quality of the output of ProbDF.** (A) False positive position below frequent itemsets boundary. (B) Precision and Recall for ProbBF.

In another figure, in Fig. 6A, all of the false positive frequent itemsets generated by the ProbDF have been shown with the real support. This figure also shows the average real support of these false positive frequent itemsets. Since these itemsets are not frequent, their support is less than the minimum support threshold (*i.e.*, minimum support count in the figure). The average support of the Chess, Connect, Pumsb, Pumsb_star, and Accidents datasets (which are mined for the support threshold of 0.5, 0.7, 0.7, 0.4, and 0.4, respectively) are 0.491, 0.699, 0.692, 0.391, and 0.39, respectively. This figure clearly indicates that these false positive frequent itemsets are closely located on the outer sphere of the frequent itemset boundary.

Lastly, we apply two widely used measures in the data mining literature, recall and precision, to evaluate the effectiveness of the ProbDF algorithm in successfully approximating all frequent itemsets. Recall represents the percentage of all true positive

frequent itemsets that are successfully predicted by ProbDF, while precision represents the percentage of probabilistic frequent itemsets generated by ProbDF that are true positive. This analysis is depicted in Fig. 6B for all four approximate algorithms compared in this study.

Two observations can be drawn from the results shown in Fig. 6B. Firstly, there are no significant differences in the quality of the results of ProbDF and ProbBF because they use the same PSPM model for their predictions. Secondly, the results of this figure closely align with those reported in *Sadeequllah et al. (2024)*. This ensures that the quality of the predictions of ProbDF is the same as that of ProbBF, albeit with more time and memory efficiency.

## Discussion of ProbDF algorithm outcomes and implications

The above findings reinforce the observations made in *Sadeequllah et al. (2024)*. It was noted in *Sadeequllah et al. (2024)* that the PSPM technique produces accurate predictions, especially when the dataset is dense and the support threshold is high. This fact is further corroborated by the results shown in Figs. 5A and 5B, which depict the outcomes of mining the Connect and PUMSB_star datasets in this study. The Connect dataset, being highly dense and mined at a support of 0.7, exhibits significantly superior results compared to the less dense PUMSB_star dataset, which is mined at a lower support of 0.4.

Additionally, Fig. 6A illustrates that false negative itemsets have probabilistic support just below the minimum support threshold. This observation aligns with the results in Fig. 5A, where most false negative frequent itemsets are included in the probabilistic frequent itemset but with one item missing.

Moreover, the results shown in this section clearly indicate that the quality of ProbDF's output deteriorates as the support threshold values are lowered. This decline is due to the scalability of the proposed methodology. As the size of the frequent itemsets increases down the search space, the error compounds, and the possibility of false outcomes rises. This compound error is also significantly influenced by the density of the dataset.

Figure 7 displays the densities of different datasets by dividing the average transaction length (after deleting the infrequent items) by the total number of frequent items. This figure shows that the Connect dataset has the highest density, while PUMSB_star has the lowest. Various studies (*Burdick et al., 2005*; *Gouda & Zaki, 2005*) report that the Average Frequent Itemset Size (AFIS) for the Connect dataset is double that of the AFIS for the PUMSB dataset at the same support threshold values. The PUMSB and Chess datasets have nearly identical AFIS, while the AFIS of the PUMSB_star dataset falls between those of PUMSB and Connect.

However, Figs. 5A, 5B, and 6B show that the quality of ProbDF's output is highest for the Connect dataset, followed by somewhat identical performances on Chess and PUMSB datasets, while the quality for PUMSB_star is the lowest. This is attributed to the highest density of the Connect dataset, followed by Chess, PUMSB, and PUMSB_star. These findings highlight the significant role that dataset density plays in the successful application of the ProbDF algorithm. Additionally, the results do not indicate that ProbDF is affected by the number of frequent items or the number of frequent itemsets in a dataset.

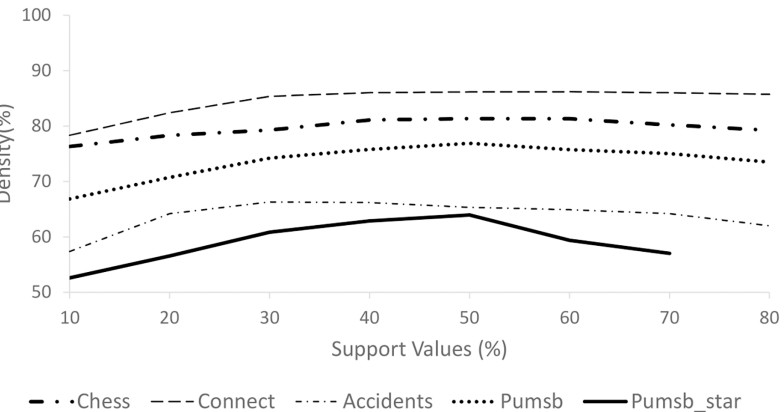

**Figure 7** **Densities of the five datasets used in comparison.**

To summarize the above discussion, the ProbDF algorithm is most effective with dense datasets, yielding higher-quality output when relatively high support threshold values are used. Additionally, ProbDF is valuable in FIM domains where some loss of quality is acceptable in exchange for increased efficiency. This is particularly advantageous in real-time environments where fast response times are prioritized over the highest quality responses. Based on the findings in this section, it is reasonable to conclude that ProbDF offers high runtime and memory efficiency with only a marginal loss in quality.

## FUTURE WORK

An important future direction is to enhance the performance of the ProbDF algorithm on certain datasets, such as PUMSB_star, where ProbDF does not perform optimally, by incorporating other approximate techniques. Another significant direction involves applying the ProbDF algorithm in real-time decision-making domains. For instance, this algorithm could be utilized in real-time business decision-making and real-time intrusion detection, where sub-optimal but rapid reponse is required. Additionally, the ProbDF algorithm could be tested on large, dense data streams where the results of other FIM algorithms designed for data streams are also approximate. ProbDF has the potential to produce more accurate results by considering all transactions and incrementally storing the support of only frequent 2-itemsets.

Furthermore, ProbDF could be extended to design a fast maximal frequent itemset mining algorithm that skips most of the computations by utilizing the PSPM model. ProbDF could also be adapted to other FIM domains, such as sequential, high-utility, parallel, and distributed FIM algorithms, to mitigate the impact of heavy computations and improve the overall efficiency of these algorithms.

## CONCLUSIONS

In summary, this article introduces ProbDF, a novel approximate frequent itemset mining algorithm designed to predict the support of itemsets. Using the PSPM technique tailored

for dense data, ProbDF efficiently forecasts the support of itemsets of size 3 or greater. Notably, ProbDF outperforms its predecessor, ProbBF, in efficiency by adopting a depth-first search strategy, significantly reducing memory usage compared to ProbBF's breadth-first search approach. This improved memory efficiency directly contributes to enhanced runtime efficiency. Additionally, ProbDF leverages promotion pruning to further optimize both memory and runtime efficiency. Furthermore, similar to its predecessor, ProbDF utilizes zero transaction data beyond two-size frequent itemsets.

ProbDF's performance was rigorously evaluated across five commonly used real-world benchmark dense datasets, where it was compared against seven state-of-the-art algorithms, including its predecessor ProbBF. Across all datasets, ProbDF consistently outperformed the competing algorithms by a significant margin, showcasing its superior efficiency. Due to its probabilistic nature, ProbDF's performance was also scrutinized regarding the quality of the results it produces. Remarkably, no significant discrepancies were observed between ProbDF and its predecessor ProbBF in terms of output quality. This investigation confirms that ProbDF achieves comparable result accuracy to ProbBF while demonstrating enhanced efficiency in both memory usage and runtime.

However, the primary limitation of the ProbDF algorithm is its dependency on the dataset's density: higher dataset density results in greater accuracy of ProbDF's outcomes. Additionally, the quality of ProbDF's output diminishes when very low support threshold values are used. To address this limitation, one potential improvement is to apply an alternative approximation method in situations where the proposed methodology produces suboptimal results. This hybrid approach could enhance the overall quality of the outcomes in scenarios where the ProbDF algorithm alone is insufficient.

### Funding
There is no external funding support for this article.

### Competing Interests
All authors of this article have no competing interests.

### Author Contributions
- Muhammad Sadeequllah conceived and designed the experiments, performed the experiments, analyzed the data, performed the computation work, prepared figures and/or tables, authored or reviewed drafts of the article, and approved the final draft.
- Azhar Rauf performed the experiments, analyzed the data, authored or reviewed drafts of the article, and approved the final draft.
- Saif Ur Rehman performed the computation work, prepared figures and/or tables, and approved the final draft.
- Noha Alnazzawi analyzed the data, prepared figures and/or tables, authored or reviewed drafts of the article, and approved the final draft.

## Data Availability

The complete JAVA source code of the proposed algorithm ProbDF is available in the Supplemental File.

The Frequent Itemset Mining Implementations Repository is available at http://fimi.ua. ac.be.

## Supplemental Information

Supplemental information for this article can be found online at http://dx.doi.org/10.7717/ peerj-cs.2334#supplemental-information.

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
