# Peer review of "Quick mining in dense data: applying probabilistic support prediction in depth-first order"

_PeerJ Computer Science, doi:10.7717/peerj-cs.2334_

## Round 0.1 · original submission · Major Revisions

Dear authors,

Thank you for submitting your article. Reviewers have now commented on your article and suggest major revisions. When submitting the revised version of your article, it will be better to address the following:

1- The research gaps and contributions should be clearly summarized in the introduction section. Please evaluate how your study is different from others in the related work section.

2- The values for the parameters of the algorithms selected for comparison should be given.

3- The paper lacks the running environment, including software and hardware. The analysis and configurations of experiments should be presented in detail for reproducibility. It is convenient for other researchers to redo your experiments and this makes your work easy acceptance. A table with parameter settings for experimental results and analysis should be included in order to clearly describe them.

4- The authors should clarify the pros and cons of the methods. What are the limitation(s) methodology(ies) adopted in this work? Please indicate practical advantages, and discuss research limitations.

5- Please include future research directions.

6- Explanation of the equations should be checked. All variables should be written in italic as in the equations. Their definitions and boundaries should be defined. Necessary references should also be given.

Best wishes,

Reviewer 1 ·

Basic reporting

- A clear description of the novelty should be added in introduction. Accordingly, revise the introduction section.
- The literature search is weak. Descriptions of frequent itemset generation needs to be added. Thus, the Introduction and/or related work section could be extended and incorporates additional discussions.

Experimental design

- The authors may consider comparing the proposed method with existing approaches.
- A proper discussion of outcomes may be added before conclusion.

Validity of the findings

- The authors should provide a complexity analysis of the proposed method.

Additional comments

- There are dot, comma, and typo errors. Many of these editing problems lie throughout the manuscript. A formal copy editing is needed.
- Future scope may be added.

Reviewer 2 ·

Basic reporting

This paper proposes a depth-first algorithm ProbDF with a search strategy. It introduces a time and space efficient single vector technique. It also introduces the concept of promotion pruning.
It is a valuable study. References are sufficient and appropriate. Table and Figures are suitable. However, major revision is required to fix some parts.

Experimental design

An analysis has been done on five benchmark datasets.
OK

Validity of the findings

The experiments demonstrate that ProbDF is efficient in terms of time and space, and successfully generates the majority of frequent itemsets on real-world benchmark datasets.
OK

Additional comments

This paper proposes a depth-first algorithm ProbDF with a search strategy. It introduces a time and space efficient single vector technique. It also introduces the concept of promotion pruning.

It is a valuable study. References are sufficient and appropriate. Table and Figures are suitable. However, the followings can be fixed:

1- Providing a table that summarizes the related work would increase the understandability of the difference from the previous studies in the "Related Works" section.

2- Some abbreviations are used in the text without giving their expansion.
For example; MAFIA, NegFIN, SPMF, RSB-PFI, etc.
The authors should write that "these abbreviations stand for what".

3- The symbols in the text should be italic.
For example:
- "ensures the support of all (k-1) itemsets of a k-itemset be already known"
- "The support of the implicit root node is set at |D| because"

4- The following phrases should be changed, since the section numbers are missing:
"In sections VI-A and VI-B,"
"Section VI-C"

5- The conclusion section can be improved. The results and main findings can be discussed.
The limitations of the proposed method should be discussed. Then, the authors can explain the possible solutions to overcome these limitations.
The authors may explain the possible future studies in the conclusion section.

6- There are many symbols. A notation table can be added to give all symbols and their meanings.

Reviewer 3 ·

Basic reporting

This paper proposes ProbDF as an improved FIM algorithm that balances efficiency and memory consumption, making it suitable for handling large, dense datasets.

The algorithms' presentation is unclear. Many equations are placed within the
algorithms, which is not ideal. Equations should be presented more clearly outside the
algorithms.

The overall presentation needs improvement.

Experimental design

While ProbDF aims to address memory issues by discarding transaction data after size two itemsets, the use of a probabilistic model might still require substantial memory, especially if the model itself or intermediate data structures are large. Clarify the same.

Validity of the findings

The probabilistic nature of ProbDF means that there might be some compromises in the quality of the results. Specifically, there might be some inaccuracies in the predicted support values, leading to false positives or false negatives in the frequent itemsets generated. Clarify the same.

Depth-first search strategies, while often more memory-efficient than breadth-first strategies, can still face challenges in very large or dense datasets. There might be situations where depth-first search does not perform optimally due to the specific structure of the data. Clarify the same.

While the study claims that ProbBF does not require additional parameters beyond the traditional support threshold, the performance of ProbDF might still be sensitive to certain implicit parameters or assumptions within the probabilistic model. These sensitivities might need careful management. Clarify the same.

As the size of frequent itemsets grows, the reliance on probabilistic support predictions might introduce scaling challenges. The study does not deeply explore how the algorithm performs as the number and size of itemsets increase significantly. Clarify the same.

---

## Round 0.2 · accepted · Accept

Dear authors,

Thank you for clearly addressing all the reviewers' comments. I confirm that the quality of your paper has been improved. In light of this revision, the paper now appears to be ready for publication.

Best wishes,

Reviewer 1 ·

Basic reporting

Authors has addressed my concerns.

Experimental design

No comment

Validity of the findings

No comment

Additional comments

No Comment

Reviewer 2 ·

Basic reporting

-

Experimental design

-

Validity of the findings

-

Additional comments

The authors revised the manuscript adequately according to the reviewers comments.
The manuscript is now more qualified and clear.
I have no further comments.
I suggest accepting it for publication in present form.